# Two New Species of *Backusella* (*Mucorales, Mucoromycota*) from Soil in an Upland Forest in Northeastern Brazil with an Identification Key of *Backusella* from the Americas

**DOI:** 10.3390/jof8101038

**Published:** 2022-09-30

**Authors:** Catarina Letícia Ferreira de Lima, Joana D’arc Alves Leitão Lundgren, Thuong Thuong Thi Nguyen, Thalline Rafhaella Leite Cordeiro, Diogo Xavier Lima, Luciana Melo Sartori Gurgel, Diogo Paes da Costa, Hyang Burm Lee, André Luiz Cabral Monteiro de Azevedo Santiago

**Affiliations:** 1Graduate Course in the Biology of Fungi, Department of Mycology, Universidade Federal de Pernambuco, Av. Prof. Moraes Rego, s/n, Recife 50740-465, PE, Brazil; 2Environmental Microbiology Laboratory C, Department of Agricultural Biological Chemistry, College of Agriculture and Life Sciences, Chonnam National University, Gwangju 61186, Korea; 3Instituto Agronômico de Pernambuco, Avenue General San Martin, 1371, Recife 50761-000, PE, Brazil; 4Microbiology and Enzymology Laboratory, Universidade Federal do Agreste de Pernambuco, Garanhuns 55292-270, PE, Brazil

**Keywords:** *Backusellaceae*, ITS rDNA, LSU rDNA, phylogeny, taxonomy

## Abstract

During a survey of *Mucorales* from a forest located in Pernambuco state, Brazil, two new *Backusella* species were discovered and described based on morphological and molecular data (internal transcribed spacer and large subunit ribosomal DNA sequences). Both species were characterized as unbranched sporangiophores and sporangia with columellae of varied shapes forming. Multispored sporangiola were frequent, whereas unispored sporangiola were rare. URM 8395 forms sporangiophores that may support hyaline, slightly curved or circinate pedicels with multispored sporangiola at their apical portion, and abundant giant cells and chlamydospores. Columellae of sporangia are hyaline, conical (majority), or ellipsoidal with a truncate base, globose to subglobose or subglobose to conical, and, rarely, with slight medial constriction. URM 8427 does not form sporangiola from pedicels, giant cells are not observed, and columellae of sporangia are globose to subglobose, cylindrical with a truncate base, some with a slight constriction, applanate, obovoid, ellipsoidal, or, rarely, conical. Some columellae may have one side more swollen than the other and some are arranged obliquely on the sporangiophores. Sterile sporangia may or may not be formed on short sporophores. The detailed description and illustration of both novel species as well as an identification key for *Backusella* from the Americas are provided.

## 1. Introduction

*Backusella* Hesseltine & J.J. Ellis belongs to the *Backusellaceae* family (established by K. Voigt and PM Kirk) and comprises fungi characterized by the production of sporangiophores that are transiently curved when young and erect when mature. These fungi possess non-apophysate and multisporate, deliquescent-walled sporangia [1], and most species form uni- and/or multispored sporangiola with a persistent wall [2]. *Backusella* species are saprobic in nature and are found in soil, litter, plant debris, herbivore dung, decaying wood, *Fragaria*, diseased roots, and *Medicago sativa* [3,4].

*Backusella* has been described as the member of the *Mucoraceae* family [5], with *B. circina* J.J. Ellis & Hesselt. as the type species of this genus. Subsequently, species of this genus were transferred to *Thamnidiaceae* Fitzp. [6], owing to their morphological similarity with *Thamnidium* Link [3]. However, Benny and Benjamin [7] monographed *Backusella* and grouped three species (*B. circina* J.J. Ellis & Hesselt., *B. ctenidia* (Durrell & M. Fleming) Pidopl. & Milko ex Benny & R.K. Benj., and *B. lamprospora* (Lendn.) Benny & R.K. Benj.) that produced sporangia and sporangiola into this genus [7].

In a study combining morphological and molecular data (internal transcribed spacer (ITS) and large subunit (LSU) of nuclear ribosomal DNA (rDNA) sequences), Walther et al. [2] transferred seven *Mucor* species that presented curved sporangiophores when young and erect at maturity to *Backusella*. In this study, *B. ctenidia* was transferred to *Mucor* (as *M. ctenidius* (Durrell & M. Fleming) Walther & de Hoog). Currently, this genus comprises 26 species, 10 of which were isolated from Australia [8], four from South Korea [3,4], and five from Brazil [9,10,11,12].

During a study on the diversity of *Mucorales* in an upland forest located in the municipalities of Arcoverde and Águas Belas, Pernambuco, Brazil, two species of *Backusella* that varied morphologically and genetically (via ITS and LSU rDNA analysis) from other species were isolated. The aim of this study was to describe and illustrate these new species. We also provide a key to the *Backusella* species from the Americas.

## 2. Materials and Methods

### 2.1. Sampling Site

Soil samples were collected from an upland forest in the city of Arcoverde (at 1000 m altitude), district of Mimoso (08°39′688″S 036°90′013″ W), and at Serra do Comunaty (09°05′441″ S 037°05′438″ W), in the municipality of Águas Belas (at 900 m altitude), both of which are located in the state of Pernambuco, Brazil. Mimoso has a tropical rainy climate with dry summers, an average temperature ranging between 12 °C and 25 °C, and an average annual rainfall of 653 mm [13,14]. The vegetation consists of deciduous and semideciduous forests. The municipality of Águas Belas also has a tropical rainy climate with dry summers and an average annual rainfall of 373 mm. The temperature ranges between 18 and 34.5 °C [15]. The vegetation is composed of semideciduous and deciduous forests [16].

### 2.2. Isolation, Purification, and Identification

Five milligrams of soil were added to wheat germ agar medium [17] containing chloramphenicol (200 mg·L^−1^) in Petri dishes in triplicate. Colony growth was monitored for 7 days at room temperature (28 ± 2 °C). To purify *Backusella* spp., mycelial fragments taken from the edge of the growing colony were transferred to malt extract agar (MEA) medium [17]. The species were identified by observing their macroscopic (appearance, color, and diameter of colonies) and microscopic (microstructures) characteristics, according to the descriptions of Benny and Benjamin (1975), Walther et al. (2013), de Souza et al. (2014), Lima et al. (2016), Crous et al. (2019) and Nguyen et al. (2021). Slides corresponding to the holotypes of the new species (URM 94839 and URM 94983) were deposited at the URM Herbarium of the Universidade Federal de Pernambuco. Ex-type living cultures of both new species were deposited at the URM Culture Collection of the Universidade Federal de Pernambuco (URM 8395 and URM 8427).

### 2.3. Experiments

Pure cultures were grown in triplicate in both MEA and potato dextrose agar (PDA) media [17] and incubated at 10, 15, 20, 25, 30, 35, and 40 °C for 15 days. For morphological identification, fragments of colonies were obtained from the plates for examination of fungal structures. These fragments were placed together with a drop of potassium hydroxide (KOH, 3%) or lactophenol blue on microscope slides and observed under a light microscope (Carl Zeiss Axioscope 40, Göttingen, Germany). Photomicrographs were captured using the Leica DM 500 microscope combined with the Leica ICC50 camera (Leica, Heerbrugg, Switzerland) and Leica Application Suite software (v.3.4). The color designation of the colonies was performed according to the terminology of Kornerup and Wanscher [18].

### 2.4. DNA Extraction, Amplification, Cloning, and Sequencing

Fungal biomass was extracted from MEA cultures in test tubes incubated at 28 °C for up to 5 days. The material was transferred to 2 mL microtubes with screw caps. To each tube, 0.5 g of acid-washed glass beads of two different diameters (150–212 μm and 425–600 μm, 1:1; Sigma, St. Louis, MO, USA) were added. The material was crushed by stirring at high speed in a FastPrep homogenizer (FastPrep-24; MP Biomedicals, Irvine, CA, USA). The genomic DNA extraction procedure was conducted as described by de Oliveira et al. [19]. Briefly, the mycelium was homogenized in cetyltrimethylammonium bromide (C TAB) lysis buffer (2% CTAB, 20 mM ethylenediaminetetraacetic acid (EDTA), 0.1 M Tris-HCl, pH 8.0, 1.4 M NaCl) [20,21], followed by washing with chloroform:isoamyl alcohol (24:1) to subsequently separate the DNA-containing supernatant from the hyphal residues. The supernatant was mixed with an equal volume of isopropanol, followed by DNA precipitation after incubation at –20 °C for 30 min. After centrifugation at 13,000 rpm for 15 min, the resulting DNA pellet was washed with 70% ethanol and resuspended in 50 μL of ultrapure water.

For amplification of the ITS and LSU rDNA, we used the primer pairs ITS1/ITS4 and LR1/LSU2 [22,23,24], respectively. Thermal cycling parameters were as follows: 5 min at 95 °C (one cycle), 45 s at 94 °C, 1 min at 60 °C, 1 min at 72 °C (39 cycles), and a final elongation of 7 min at 72 °C. The final amplicons were purified with the enzymatic mix NucleoSAP (Cellco Biotech of Brazil, São Carlos, Brazil) and used for sequencing at the Genomic Technology and Gene Expression Platform of the Center for Biological Sciences of the Federal University of Pernambuco-UFPE (Pernambuco, Brazil). Direct sequencing of the ITS region from PCR products of strain URM 8427 failed. PCR products were cloned using the pGEM-T Easy Vector (Promega, Madison, WI, USA), following the manufacturer’s instructions. These clones were sequenced using the primers M13F forward (5′-GTAAAACGACGGCCAGT-3′) and M13R reverse (5′-GCGGATAACAATTTCACACAGG-3′).

Sequence data were compared with similar sequences available in the National Center for Biotechnology Information GenBank database using Nucleotide BLAST (BLASTn). The newly obtained sequences were deposited in the GenBank databases (Table 1).

### 2.5. Phylogenetic Analyses

The phylogenetic relationship of the new *Backusella* species and related species was determined by analysis of concatenated sequence datasets of two loci (ITS and LSU). Sequences were aligned via multiple alignment using Fast Fourier Transform (MAFFT) v.7 (https://mafft.cbrc.jp/alignment/server, accessed on 21 July 2022) [25,26] and manually improved in Molecular Evolutionary Genetics Analysis (MEGA) v.7 [27]. The analysis of the concatenated sequence datasets of the two loci was performed in MEGA v.7. Maximum likelihood (ML) analysis was performed in Randomized Axelerated Maximum Likelihood (RAxML)-HPC v. 8.2.8 Black Box [28] using the general time-reversible (GTR)+G+I model test with 1000 bootstrap (BS) replicates on the CIPRES Science Gateway Portal (https://www.phylo.org/portal2/) [29]. Bayesian inference (BI) was performed using MrBayes v. 3.2.2 [30] on The Extreme Science and Engineering Discovery Environment (XSEDE) through CIPRES using the GTR+I+G model as selected by jModelTest v.2.1.10 under the Akaike information criterion for BI [31,32]. BI analysis was conducted with 1 × 10^6^ generations with a burning value of 25%. Phylogenetic trees were viewed and arranged using Interactive Tree of Life v4 (https://itol.embl.de, accessed on 21 July 2022) [33]. Values less than 0.90 BI posterior probability (BPP) and 70% ML bootstrap (MLBS) are not shown. Sequence data were compared with those of similar sequences available in the GenBank database using BLASTn.

## 3. Results

### 3.1. Phylogenetic Analyses

Phylogenetic analyses of the combined ITS and LSU sequence data identified 53 taxa, including the newly generated sequences and outgroup (Figure 1). The sequence dataset consisted of 1828 characters including gaps (ITS, 1098 characters; LSU, 730 characters). Results obtained using BI and maximum likelihood analyses phylogenetically characterized URM 8395 and URM 8427 as new species that forms a distinct clade from other *Backusella* species with 100% MLBS and 1 BPP support values (Figure 1). URM 8395 is close to *B. dispersa* and to a clade containing URM 8427, *B*. *azygospora, B. mclennaniae, B. liffmaniae, B. psychrophila, B. lamprospora*, and *Backusella* as ‘group X’. URM 8427 is closer to the not yet described *Backusella* ‘group X’.

### 3.2. Taxonomy

*Backusella brasiliensis* C.L. Lima, Hyang B. Lee & A.L. Santiago, sp. nov. (Figure 2, Table 2).

Index Fungorum: IF559794;

Etymology: *brasiliensis*, referring to the country where the species was first isolated.

Colony with cottony appearance, initially white and then becoming pale gray (1–1B), colonizing the entire Petri dish (9 cm diameter) after incubation for 4 days on PDA at 25 °C; reverse yellowish-gray (2–2B). Rhizoids well-branched. Sporangiophores occasionally with a slight constriction below the sporangia, hyaline, arising directly from the substrate, recurved when young and erect at maturity, up to 12 µm diameter, unbranched or, rarely, with simple branches. Sterile branches common on long sporangiophores. Both sporangiophores and branches have multispored sporangium at their apex, with a smooth to slightly encrusted wall. One septum is commonly formed near the branching point. The main axes of sporangiophores may support hyaline, slightly curved or circinate pedicels (up to 95 × 45 μm^2^) with multispored sporangiola at their apical portion and a smooth to slightly encrusted outer wall. Sporangia yellowish, globose to subglobose (up to 60–70 µm diameter), wall deliquescent and slightly echinulate. Columellae of sporangia hyaline, conical (majority), or ellipsoidal with truncate base, globose to subglobose or subglobose to conical, and, rarely, with a slight constriction at center, (12–) 15–25 (–30) × (15–) 17–30 µm; wall lightly encrusted; collar evident. Short, simple or sympodially branched (up to seven times) sporophores bearing only multispored and/or unispored sporangiola (rare) formed near the substrate. Sporangiola hyaline, globose to subglobose, (12–) 15–45 (–50) μm diameter, echiinulate, and with a persistent wall; columellae of sporangiola globose to subglobose or subglobose to conical, 7–15 µm diameter, hyaline, and with a slightly encrusted wall. Sporangiospores hyaline, globose to subglobose (majority), 4–12 (–17) μm diameter, or irregular, 20–30 × 12–17 μm, smooth-walled. Abundant giant cells observed, hyaline, some with yellowish content, 7–17 µm diameter. Chlamydospores abundant, hyaline, 7–35 × 9.5–17 µm, wall smooth to slightly encrusted. No zygospores were observed.

Habitat: Soil

Distribution: Pernambuco, Brazil.

Specimen examined: BRAZIL, Pernambuco, Mimoso (08°39′688″ S 036°90′013″ W), soil, 2019, C.L.F. Lima (holotype URM 94839, ex-type URM 8395).

Media and temperature tests: On PDA, at 10 °C, slow growth (3 cm diameter in 144 h) with poor sporulation; at 15 °C, better growth (6 cm in 120 h) than that at 10 °C with good sporulation; at 20 °C, good growth (7 cm in 120 h) with good sporulation; at 25 °C, better growth (9 cm in 96 h) with good sporulation; at 30 °C, good growth (9 cm in 144 h) and good sporulation; at 35 °C, slow growth (2 cm in 120 h) with lack of sporulation; and at 40 °C, lack of growth. A similar growth pattern was observed on MEA and PDA at all temperatures.

*Backusella obliqua* C.L. Lima, J.D. Leitão, Hyang B. Lee & A.L. Santiago, sp. nov. (Figure 3, Table 2).

Index Fungorum: IF559793;

Etymology: *obliqua*, referring to the columellae arranged obliquely on the sporangiophores.

Colony cottony, white (1–1A), colonizing the entire Petri dish (9 cm diameter) after 4 days on PDA at 25 °C; reverse grayish white (1–1B). Rhizoid-like sterile branched hyphae common. Sporangiophores terminating in a multispored sporangium at their apex, hyaline or with greenish contents, occasionally with slight constriction below the sporangia, arising directly from the substrate, recurved when young and erect at maturity, simple or sympodially branched up to three times, up to 12 µm diameter. A septum is commonly formed near the branch point. Sporangia greenish-yellow, globose to subglobose, 15–60 µm diameter, wall slightly echinulate, deliquescent. Columellae of sporangia hyaline or with greenish-yellow contents, globose to subglobose, cylindrical with a truncate base, some with a slight constriction, applanate, obovoid, ellipsoidal, or, rarely, conical, (9.5–) 12–35 × 15–30 µm, smooth walled; collar little evident when present. Some columellae may have one side more swollen than the other and some are arranged obliquely on the sporangiophores. Short sporophores commonly sympodially branched, up to nine times, containing multispored and/or, rarely, unispored sporangiola (some sterile), are formed close to the substrate. Sporangiola greenish gray, globose to subglobose (9–) 12–24 μm diameter, wall echinulate and persistent. Columellae of sporangiola hyaline, globose to subglobose and applanate, 7–12 µm diameter, wall slightly encrusted. Sporangiospores hyaline or with greenish-yellow contents, globose to subglobose 7–12 (–15) μm diameter, smooth-walled. Giant cells and zygospores not observed.

Habitat: Soil

Distribution: Pernambuco, Brazil.

Specimen examined: BRAZIL, Pernambuco, Águas Belas (09°05′441″ S 037°05′438″ W), soil, 2019, C.L.F. Lima (holotype URM 94983, ex-type URM 8427).

Media and temperature tests: On PDA, at 10 °C, slow growth (2 cm in 120 h) with poor sporulation; at 15 °C, better growth (5 cm in 120 h) than that at 10 °C with good sporulation; at 20 °C, good growth (7 cm in 120 h) with good sporulation; at 25 °C, better growth (9 cm in 96 h) with good sporulation; at 30 °C, good growth (9 cm in 144 h) and good sporulation; at 35 °C, slow growth (3 cm in 120 h) with lack of sporulation; and at 40 °C, lack of growth. A similar growth pattern was observed on MEA and PDA at all temperatures.

## 4. Discussion

*Backusella brasiliensis* (URM 8395) and *B. obliqua* (URM 8427) were described in this study based on their morphological and molecular characteristics that make them different from the other closely related species; therefore, they were noted as new species. Phylogenetically, *B. brasiliensis* formed a clade distant from other *Backusella* species but closer to *B. dispersa* and to a clade containing URM 8427, *B*. *azygospora, B. mclennaniae, B. liffmaniae, B. psychrophila, B. lamprospora* and *Backusella* as ‘group X’ (Figure 1). However, these species are morphologically different. *Backusella brasiliensis* is differentiated from *B. azygospora* as it does not produce azygospores, which are common in the later species. Besides, chlamydospores and sterile branches in long sporangiophores are not observed in *B. azygospora* [11]. *Backusella dispersa* forms smaller sporangia (30.2–47.5 × 28.3–46.1 μm) and larger columellae (19.5–38.7 × 18.5–33.4 μm) than those observed for the new species. In addition, sporangiospores of *B. dispersa* are globose to broadly ellipsoidal (8–12 × 7–10 μm), and are different from those of the new species, which are globose to subglobose (majority), 4–12 (–17) μm diameter or irregular, 20–30 × 12–17 μm. Although no chlamydospores are formed in *B. dispersa* [8], they are common in URM 8395. The new species is easily differentiated from *B*. *mclennaniae*, *B. liffmaniae* and *B*. *psychrophila* by forming giant cells and chlamydospores. Yet, *B*. *brasiliensis* forms columellae that are conical (majority), ellipsoidal with truncate base, globose to subglobose or subglobose to conical, and, rarely, with a slight constriction at center, whereas columellae of *B*. *mclennaniae* and *B. liffmaniae* and *B*. *psychrophila* are globose, ellipsoidal or applanate.

*Backusella brasiliensis* and *B. lamprospora* present morphological similarities, such as the following: the main axes of their sporangiophores support hyaline, slightly curved or circinate pedicels with multispored sporangiola at their apical portion; short and simple or sympodially branched sporophores bearing only multispored and/or unispored sporangiola (rare) are formed near the substrate [7]. However, the main sporangiophore axes of *B. lamprospora* are simple or sympodially branched, differing from those of the new species that are unbranched or once-branched. *Backusella lamprospora* is characterized by forming sporangiophores with hemispherical to globose or ovoid columellae, differing from those found in *B. brasiliensis*, which are conical, ellipsoidal with a truncate base, globose to subglobose, subglobose to conical, conical, or cylindrical, including some with a slight constriction at the center. Furthermore, the sporangiospores of *B. brasiliensis* are globose, subglobose, and irregular, whereas those of *B. lamprospora* are only subglobose [7].

The formation of giant cells and chlamydospores is not common in *Backusella* species, and as far as we know, in addition to the new species, giant cells have been cited only in *B. gigacellularis* and *B. oblongielliptica*, whereas chlamydospores have only been observed in *B. chlamydospora*. *Backusella gigacellularis* does not produce unispored sporangiola, and its sporangia are larger (40–131 μm diameter) than those of *B. brasiliensis*. Moreover, sporangiospores of the new species are globose, subglobose, and irregular, whereas those of *B. gigacellularis* are ellipsoidal and, rarely, irregular [9]. *Backusella oblongielliptica* can be easily differentiated from *B. brasiliensis* by the lack of chlamydospore and sporangiola formation and the production of oblongly ellipsoidal sporangiospores [2,34]. *Backusella chlamydospora* differs from *B. brasiliensis* as it produces abundant unispored sporangiola, larger sporangia (35–80 × 35–75 μm), and subglobose, conical, ellipsoidal, cylindrical, hemispherical, or nearly pyriform columellae, which are sometimes bell-shaped, long conical, or constricted at the center [4]. In contrast, *B. brasiliensis* rarely produces unispored sporangiola and does not form cylindrical or pyriform columellae.

Phylogenetically, *B. obliqua* (URM 8427) is close to the *Backusella* species of ‘group X’, which is a group of a putative new species not yet proposed [8], but also close to *B*. *lamprospora* and *B*. *psychrophila.* Morphologically, *Backusella* ‘group X’ forms sporangiophores up to 8 µm diameter, unlike *B. obliqua*, which forms sporangiophores up to 12 µm diameter. The columellae produced by *B. obliqua* are globose to subglobose, subglobose to irregular, cylindrical with a truncate base (some with slight constriction), flattened, obovoid, ovoid, ellipsoidal, or, rarely, conical, some arranged obliquely on the sporangiophores, and some with one side more swollen than the other, while *Backusella* ‘group X’ forms only globose, ellipsoidal, or applanate columellae. The sporangiospores of *B. obliqua* are globose to subglobose, unlike that of *Backusella* ‘group X’, which forms globose to broadly ellipsoidal sporangiospores. *Backusella obliqua* can be differentiated from *B*. *lamprospora* by forming sporangiophores without lateral pedicellate sporangiola. Columellae of *B*. *lamprospora* are hemispherical to globose or ovoid, differing from those of *B*. *obliqua* [7]. This new species forms sporangiophores that are simple or sympodially branched up to three times, whereas sporangiophores of *B*. *psychrophila* are occasionally branched. In addition, sporangiospores of *B*. *obliqua* are globose to subglobose, whereas those of *B*. *psychrophila* are broadly ellipsoidal to ellipsoidal [8].

In conclusion, *B. obliqua* and *B. brasiliensis* are morphologically and genetically different from other *Backusella* species, justifying their recognition as novel species. Brazil is a tropical country with areas considered global biodiversity hotspots that support high fungal diversity [12,35,36,37]. Voigt et al. [38] showed that of the 74 *Mucoromycota* species newly described between 2015 and 2020, 54 (74%) were isolated from Brazil, South Korea, Australia, and China, with 17 new species described in Brazil, confirming the great potential for the discovery of new taxa of *Mucoromycota* in this country. Regarding *Backusella*, of the 26 accepted species, only six *(B. azygospora, B. brasiliensis, B. constricta, B. gigacellularis, B. lamprospora*, and *B. variabilis*) were isolated from Brazil, including from areas of upland forest [10,11,12,39]. Considering that a high diversity of fungi is expected in tropical hotspots, we believe that this number of known *Backusella* species in Brazil is underestimated and it could increase with more investments in research and further training of Brazilian taxonomists studying *Mucoromycota*.

## Figures and Tables

**Figure 1 jof-08-01038-f001:**
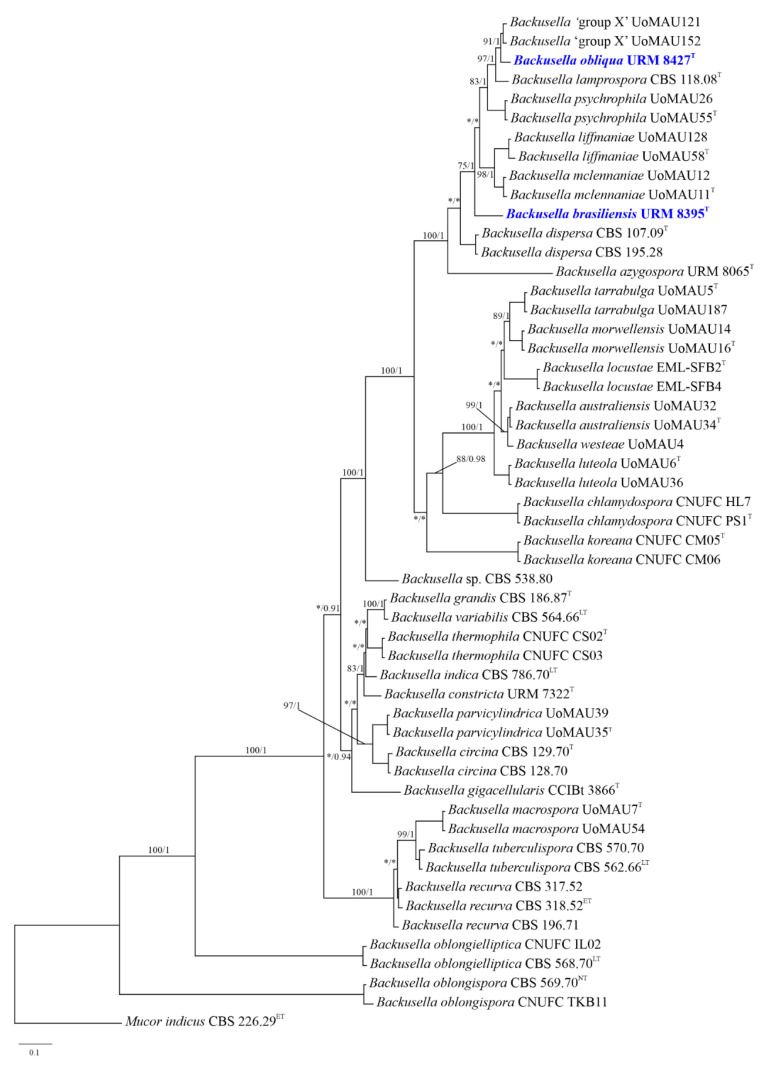
Phylogenetic tree of *Backusella* based on maximum likelihood analysis of a combined DNA dataset of internal transcribed spacer and large subunit sequences. Bootstrap values for Bayesian posterior probabilities over 0.90 and maximum likelihood higher than 70% are placed above the branches. Bootstrap values lower than 0.90 and 70% are marked with “*”. The bar indicates the number of substitutions per position. *Mucor indicus* CBS 226.29 was used as an outgroup. The newly generated sequences of *B*. *obliqua* URM 8427 and *B. brasiliensis* URM 8395 are indicated in blue. Ex-type, ex-epitype, ex-lectotype, and ex-neotype strains are marked with T, ET, LT, and NT, respectively.

**Figure 2 jof-08-01038-f002:**
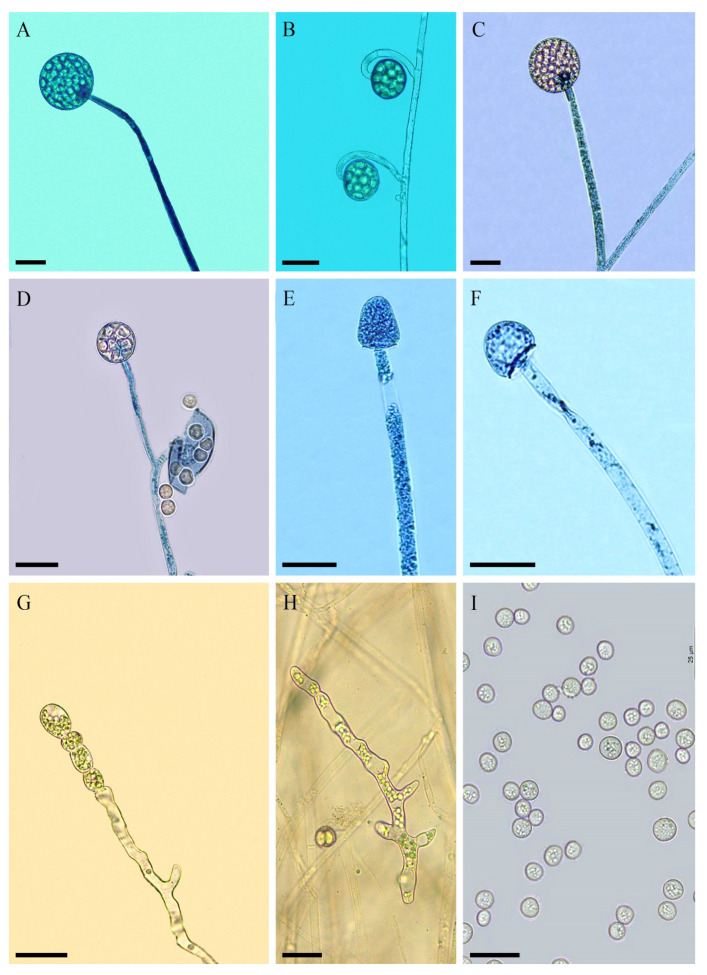
*Backusella brasiliensis* (URM 8395): (**A**) sporangiophore with sporangium; (**B**) pedicellate sporangiola arising from the sporangiophore; (**C**) sporangiophore branch with sporangium; (**D**) short-branched sporophore bearing only sporangiola; (**E**,**F**) sporangiophore with columella; (**G**) terminal chlamydospores; (**H**) giant cell; and (**I**) sporangiospores. Scale bars = 25 μm.

**Figure 3 jof-08-01038-f003:**
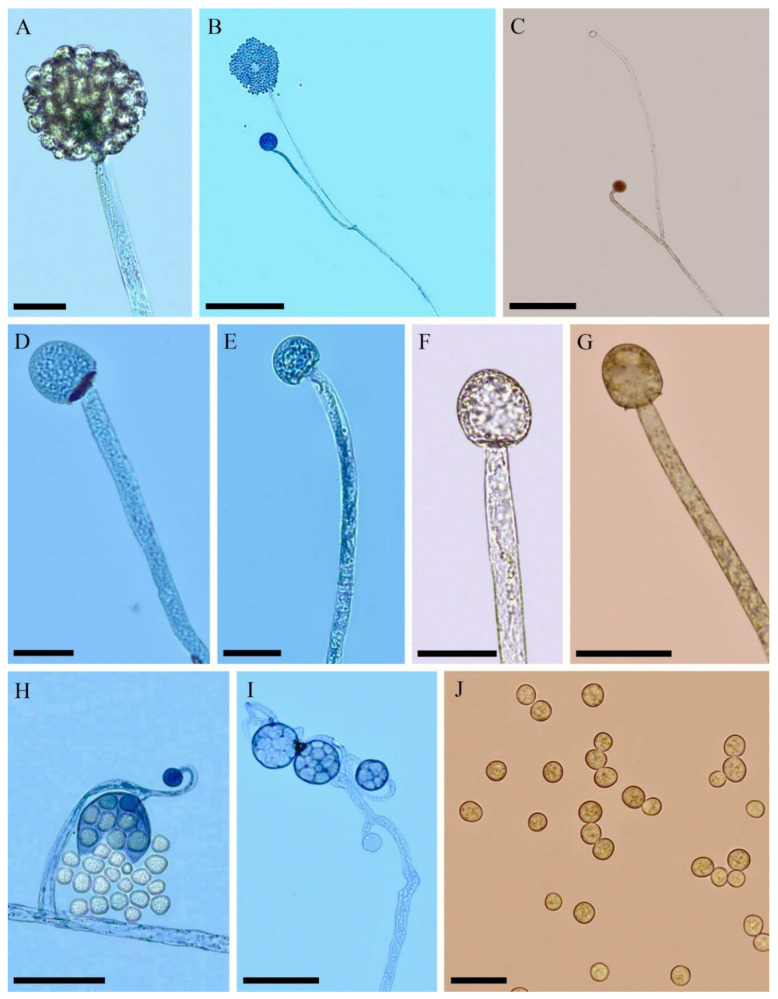
*Backusella obliqua* (URM 8427): (**A**) sporangiophore with sporangium; (**B**) branched sporangiophore with sporangia; (**C**) branched sporangiophore with sporangium and columella; (**D****–G**) sporangiophore with columella; (**H**) short-branched sporophore bearing uni- and multispored sporangiola; (**I**) short-branched sporophore bearing only multispored sporangiola and columella; and (**J**) sporangiospores. Scale bars: (**A**,**D**–**G**,**J**) = 25 μm; (**B**,**C**) = 200 μm; (**H**,**I**) = 50 μm.

**Table 1 jof-08-01038-t001:** Culture collection accession numbers and voucher numbers of sequences used for the phylogenetic analyses.

Taxon Name	Collection No.	Country	GenBank Accession Number	
			ITS	LSU
*Backusella australiensis*	UoMAU34	Australia	MK959062	MK958800
*Backusella australiensis*	UoMAU32	Australia	-	MK958802
*Backusella azygospora*	URM 8065	Brazil	MK625216	MK625222
** *Backusella brasiliensis* **	**URM 8395**	**Brazil**	**OM458082.1**	**OM458083.1**
*Backusella chlamydospora*	CNUFC PS1	South Korea	MZ171385	MZ148709
*Backusella chlamydospora*	CNUFC HL7	South Korea	MZ171386	MZ148710
*Backusella circina*	CBS 128.70	USA	JN206258	JN206529
*Backusella circina*	CBS 129.70	USA	JN206257	MH871299
*Backusella constricta*	URM 7322	Brazil	KT937158	KT937156
*Backusella dispersa*	CBS 107.09	Norway	JN206269	MH866118
*Backusella dispersa*	CBS 195.28	USA	JN206271	JN206530
*Backusella gigacellularis*	CCIBt 3866	Brazil	KF742415	-
*Backusella grandis*	CBS 186.87	India	JN206252	JN206527
*Backusella indica*	CBS 786.70	India	JN206255	MH871743
*Backusella koreana*	CNUFC CM05	Korea	MZ171387	MZ148711
*Backusella koreana*	CNUFC CM06	Korea	MZ171388	MZ148712
*Backusella lamprospora*	CBS 118.08	Switzerland	NR_145291	NG_058650
*Backusella liffmaniae*	UoMAU128	Australia	-	MK958735
*Backusella liffmaniae*	UoMAU58	Australia	MK959065	MK958734
*Backusella luteola*	UoMAU6	Australia	MK959058	MK958795
*Backusella luteola*	UoMAU36	Australia	-	MK958794
*Backusella locustae*	EML-SFB4	Korea	KY449293	KY449290
*Backusella locustae*	EML-SFB2	Korea	KY449291	KY449292
*Backusella macrospora*	UoMAU7	Australia	MK959107	MK958628
*Backusella macrospora*	UoMAU54	Australia	-	MK958629
*Backusella mclennaniae*	UoMAU12	Australia	MK959087	MK958777
*Backusella mclennaniae*	UoMAU11	Australia	MK959077	MK958776
*Backusella morwellensis*	UoMAU16	Australia	MK959059	MK958808
*Backusella morwellensis*	UoMAU14	Australia	-	MK958806
** *Backusella obliqua* **	**URM 8427**	**Brazil**	**ON858475**	**ON858467**
*Backusella oblongielliptica*	CBS 568.70	Japan	JN206278	JN206533
*Backusella oblongielliptica*	CNUFC IL02	Korea	MZ171391	MZ148715
*Backusella oblongispora*	CBS 569.70	Japan	JN206251	JN206407
*Backusella oblongispora*	CNUFC TKB11	Korea	MZ420786	MZ148717
*Backusella parvicylindrica*	UoMAU35	Australia	MK959109	MK958727
*Backusella parvicylindrica*	UoMAU39	Australia	-	MK958728
*Backusella psychrophila*	UoMAU26	Australia	-	MK958748
*Backusella psychrophila*	UoMAU55	Australia	MK959093	MK958749
*Backusella recurva*	CBS 318.52	USA	JN206261	JN206522
*Backusella recurva*	CBS 317.52	Macedonia	JN206262	MH868593
*Backusella recurva*	CBS 196.71	-	JN206265	JN206523
*Backusella tarrabulga*	UoMAU187	Australia	-	MK958805
*Backusella tarrabulga*	UoMAU5	Australia	MK959060	MK958804
*Backusella thermophila*	CNUFC CS02	Korea	MZ171389	MZ148713
*Backusella thermophila*	CNUFC CS03	Korea	MZ171390	MZ148714
*Backusella tuberculispora*	CBS 562.66	India	JN206267	JN206525
*Backusella tuberculispora*	CBS 570.70	Japan	JN206266	MH871631
*Backusella variabilis*	CBS 564.66	India	JN206254	JN206528
*Backusella westeae*	UoMAU4	Australia	MK959061	MK958796
*Backusella* sp.	CBS 538.80	Egypt	HM999964	HM849692
*Backusella* ‘group X’	UoMAU121	Australia	MK959103	MK958792
*Backusella* ‘group X’	UoMAU152	Australia	MK959102	MK958791
*Mucor indicus*	CBS 226.29	Switzerland	MH855050	HM849690

The currently isolated species and accession numbers determined in the study are indicated in bold. CBS: Culture collection of the Westerdijk Fungal Biodiversity Institute, The Netherlands; CCIBt: Collection of Algae, Cyanobacteria, and Fungi Cultures of the Botany Institute; URM: URM Culture Collection, Universidade Federal de Pernambuco, Recife, Brazil; U0MAU: National Herbarium of Victoria, Australia; CNUFC: Chonnam National University Fungal Collection, Gwangju, South Korea; EML: Environmental Microbiology Laboratory, Fungarium, Chonnam National University, Gwangju, South Korea; ITS: internal transcribed spacer rDNA; LSU: large subunit rDNA.

**Table 2 jof-08-01038-t002:** Identification key for species of *Backusella* in the Americas.

1. Sporangiola formed	2
1. Sporangiola not formed	*B. oblongielliptica*
2. Unispored sporangiola abundant	*B. circina*
2. Unispored sporangiola rare or not formed	3
3. Giant cells formed	4
3. Giant cells not formed	5
4. Columellae of sporangia ellipsoidal, cylindrical, rarely pyriform; chlamydospores absent	*B. gigacellularis*
4. Columellae of sporangia conical (majority), but also ellipsoidal with a truncate base, globose to subglobose, subglobose to conical, or, rarely, conical or cylindrical with slight constriction at the center; chlamydospores abundant	*B. brasiliensis*
5. Azygospores formed	*B. azygospora*
5. Azygospores not formed	6
6. Sporangiospores ellipsoidal	7
6. Sporangiospores not ellipsoidal	8
7. Sporangia up to 150 (–200) μm diam.; sporangiospores 20–26 × 10–12 μm	*B. recurva*
7. Sporangia up to 100 (–125) μm diam.; sporangiospores 11–15 × 7–9 μm	*B. variabilis*
8. Sporangiophores forming a terminal sporangium and few lateral pedicellate sporangiola	*B*. *lamprospora*
8. Sporangiophores forming a terminal sporangium with no lateral pedicellate sporangiola	9
9. Columellae of sporangia of varied shapes, some arranged obliquely on the sporangiophores, some with one side more swollen than the other; sporangiospores globose to subglobose	*B. obliqua*
Columellae of sporangia conical and cylindrical, sometimes constricted at the center, never arranged obliquely on the sporangiophores nor with one side more swollen than the other; sporangiospores subglobose to ellipsoidal, some slightly irregular	*B. constricta*

## Data Availability

The ITS and LSU rDNA sequences generated during the current study were submitted to Genbank (https://www.ncbi.nlm.nih.gov). The newly described species were deposited in Index Fungorum (http://www.indexfungorum.org).

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
