# Peer review of "Two New Species of Backusella (Mucorales, Mucoromycota) from Soil in an Upland Forest in Northeastern Brazil with an Identification Key of Backusella from the Americas"

_jof, 2022, doi:10.3390/jof8101038_

Round 1
Reviewer 1 Report
Two new species of Backusella (Mucorales, Mucoromycota) from soil in an upland forest in northeastern Brazil with an identification key of Backusella from America
General comments:
It is good piece of taxonomic work based on multiple markers, it is being recommended with some corrections. English needs some revisions.
Abstract:
Line 19, as unbranched sporangiophores forming
Key words:
Reduce the key words:
Remove “two new species” should be replaced with another suitable keyword
Introduction:
English need some revisions. Sentence making can be improved/
Lin 41
Backusella was described in 1969 as belonging to the Mucoraceae family
Re write like.
Backusella has been described as the member of the Mucoraceae family (Ref. 1969)
Line 42
type strain for this genus
re-write like
Strain is totally a wrong word
Type species of this genus or only type of this genus.
Introduction is lacking the introduction of
Backusellaceae, must be mentioned.
Materials and Methods
Line line77-82 need a proper reference
DNA extraction is also without reference.
Line 120-121 so require a reference.
Results
The legend of Figure 1 should be proper labeled with the names of new species mentioned these are blue colored in the tree.
Discussion:
Discussion also be improved showing the differences of your species with other closely related species in the light of literature. English should be improved.
Line 314-15 molecular characteristics that differentiated them from the other species; not making sense.
molecular characteristics that make them different from the other closely related species;
Your results must be discussed with closely related species to your both species which are: B. psychrophila, B.liffimaniae, B.mclennae from your tree.then the authors can conclude about something about new species.
Author Response
The authors would like to thank the reviewer for the excellent work. Most suggestions and corrections have been accepted and we consider that the manuscript has improved significantly now. At some instances we were not in agreement with specific criticisms that were made. For these we are providing a specific rebuttal reply below. The manuscript was revised by a native English speaker. We have uploaded the manuscript with corrections as track changes.
1 - Introduction is lacking the introduction of Backusellaceae, must be mentioned.
R. We have mentioned Backusellaceae in the first line of introduction.
2 - Line 120-121 so require a reference.
R. We believe it is not necessary to add any reference for this methodology, because we just followed the manufacturer’s instructions.

Reviewer 2 Report
Dear author,
Overall good. Is there are any possibility to improve the photo plates, specially sporangiospores photos.
Thank you
Author Response
The authors would like to thank the reviewer for the excellent work.
We believe our plates are in good quality, and sporangiospores easily observed, thus we prefer not to change them. We have improved our manuscript by correcting some weakness as track changes.
With best wishes,
André
With best wishes,
André
